# Development of a Quadruplex RT-qPCR for the Detection of Porcine Astrovirus, Porcine Sapovirus, Porcine Norovirus, and Porcine Rotavirus A

**DOI:** 10.3390/pathogens13121052

**Published:** 2024-11-29

**Authors:** Junxian He, Kaichuang Shi, Yuwen Shi, Yanwen Yin, Shuping Feng, Feng Long, Sujie Qu, Xingju Song

**Affiliations:** 1College of Animal Science and Technology, Guangxi University, Nanning 530005, China; ces5734@yeah.net (J.H.); shiyuwen2@126.com (Y.S.); 2Guangxi Center for Animal Disease Control and Prevention, Nanning 530001, China; yanwen0349@126.com (Y.Y.); fsp166@163.com (S.F.); longfeng1136@163.com (F.L.); mingdao120@126.com (S.Q.)

**Keywords:** porcine astrovirus, porcine sapovirus, porcine norovirus, porcine rotavirus species A, multiplex RT-qPCR

## Abstract

Porcine astrovirus (PoAstV), porcine sapovirus (PoSaV), porcine norovirus (PoNoV), and porcine rotavirus A (PoRVA) are newly discovered important porcine diarrhea viruses with a wide range of hosts and zoonotic potential, and their co-infections are often found in pig herds. In this study, the specific primers and probes were designed targeting the ORF1 gene of PoAstV, PoSaV, and PoNoV, and the VP6 gene of PoRVA. The recombinant standard plasmids were constructed, the reaction conditions (concentration of primers and probes, annealing temperature, and reaction cycle) were optimized, and the specificity, sensitivity, and reproducibility were analyzed to establish a quadruplex real-time quantitative RT-PCR (RT-qPCR) assay for the detection of these four diarrheal viruses. The results demonstrated that the assay effectively tested PoAstV, PoSaV, PoNoV, and PoRVA without cross-reactivity with other swine viruses, and had limits of detection (LODs) of 138.001, 135.167, 140.732, and 132.199 (copies/reaction) for PoAstV, PoSaV, PoNoV, and PoRVA, respectively, exhibiting high specificity and sensitivity. Additionally, it displayed good reproducibility, with coefficients of variation (CVs) of 0.09–1.24% for intra-assay and 0.08–1.03% for inter-assay. The 1578 clinical fecal samples from 14 cities in Guangxi Province, China, were analyzed via the developed assay. The results indicated that the clinical samples from Guangxi Province exhibited the prevalence of PoAstV (35.93%, 567/1578), PoSaV (8.37%, 132/1578), PoNoV (2.98%, 47/1578), and PoRVA (14.32%, 226/1578), and had a notable incidence of mixed infections of 18.31% (289/1578). Simultaneously, the 1578 clinical samples were analyzed with the previously established assays, and the coincidence rates of these two approaches exceeded 99.43%. This study developed an efficient and precise diagnostic method for the detection and differentiation of PoAstV, PoSaV, PoNoV, and PoRVA, enabling the successful diagnosis of these four diseases.

## 1. Introduction

Astrovirus is a member of the *astroviridae* family, which consists of 7 Astroviruses and 33 Mamastroviruses [1]. Astroviruses are widely distributed throughout the world and are infectious to 6 species of poultry, including duck and turkey, and 31 species of mammals, including humans, dog, cat, pig, sheep, and cattle [2]. Porcine astrovirus (PoAstV) is a vesicular, single-stranded, positive-sense RNA virus that is classified into the *Mamastrovirus* genus. The viral genome has three open reading frames (ORFs) consisting of ORF1a, ORF1b, and ORF2, of which ORF2 encodes phage proteins, and ORF1a and ORF1b encode nonstructural proteins and RNA-dependent RNA polymerase (RdRp) [3]. PoAstV can be categorized into five genotypes based on the ORF2 gene, including PoAstV1, PoAstV2, PoAstV3, PoAstV4, and PoAstV5 [1]. PoAstV was first found in 1980 in piglet diarrheic samples by electron microscopy. It was reported that the positivity rates of PoAstV were found to be 28.91%, 31.3%, 6.03%, 41.04%, 30.5%, and 32.3% in East, South, North, Central, Northeast, and Southwest China, respectively [4]. PoAstV usually co-infects with other diarrhea-associated viruses such as coronaviruses and sapeloviruses, leading to reduced piglet survival and serious damage to the pig industry [5].

Sapovirus (SaV) is a member of the *Sapovirus* genus in the *Caliciviridae* family, capable of causing gastroenteritis in humans as well as animals [6]. The disease has a global distribution, and the virus has been found in animals such as pigs, dogs, and cattles [7]. In 1980, porcine sapovirus (PoSaV) was discovered in piglet fecal samples using an electron microscope [8]. SaV is a non-envelope, small virus with a diameter of 23–32 nm, and has an icosahedral symmetric capsid with most of the proteins folded to form dimers, each of which forms an arched capsid [9]. The SaV genome is a single-stranded positive-sense RNA, with a total length of 7–8 kb. Based on the nucleotide sequence of the viral capsid protein, SaV is classified into 19 gene groups, of which groups GI, GII, GIV, and GV infect humans, while group GIII mainly infects pig [10]. PoSaV has a global distribution and has been reported in many countries such as the United States, Japan, the Netherlands, Canada, the United Kingdom, Italy, and South Korea [9,11]. In China, PoSaV was first reported in piglets with outbreaks of diarrhea in 2008 [12].

Norovirus is a member of the genus *Norovirus* of the family *Caliciviridae*, and has become an important pathogen of non-bacterial diarrhea in different countries [13]. Norovirus is a single-stranded positive-sense RNA virus with a diameter of 26–35 nm, and a genome size of about 7.7 kb, consisting of three ORFs [14]. According to the nucleocapsid and RNA polymerase genes, noroviruses can be classified into five genetic groups (GI–GV), and porcine norovirus (PoNoV) belongs to the GII group [15,16]. In 1997, PoNoV was first detected in the feces of healthy pigs in Japan and identified as group GII [17]. The virus has since been detected in pigs in other countries. PoNoVs were usually detected in asymptomatic adult pigs [18]. However, group GII PoNoV has been detected in both symptomatic and subclinical pigs from the United States, Latin America, and Europe [19]. Norovirus has become one of the most important pathogens of public health concern due to its environmental adaptability, diversity of genetic groups, carriage by crypto-infected individuals, and multiple vectors of transmission [20].

Rotavirus (RV) belongs to the genus *Rotavirus* in the family *Reoviridae* [21], and is responsible for gastroenteritis in several hosts, including mammals and birds [22]. There are eleven segments of double-stranded RNA that make up the genome of rotavirus. These segments encode six structural proteins (VP1–4, VP6, and VP7) and six nonstructural proteins (NSP1–6) [23]. Porcine rotavirus (PoRV) can be classified into A, B, C, and H species. It is endemic in swine herds, with varying infection rates reported worldwide [21,24,25]. It has been shown that both PoRVA and PoRVC are associated with diarrhea in piglets and weaned piglets. However, PoRVA is a major rotavirus with a high infection rate across a variety of herds. Conversely, PoRVB is more commonly associated with older pigs [25]. PoRVA was initially identified in diarrheic piglets in Australia in 1975 [26], and at present PoRVA has a global distribution [27].

Pigs infected with PoAstV, PoSaV, PoNoV, and PoRVA usually show similar clinical signs and pathological changes of gastroenteritis, and it is hard to distinguish them. Co-infection by these viruses increases the difficulty of differential diagnosis [5,6,28]. Therefore, it is relied on laboratory testing techniques for differential diagnosis, which promotes the prevention and control of diarrheal diseases. Real-time quantitative PCR/RT-PCR (qPCR/RT-qPCR) is extensively utilized for detecting various diarrheal viruses, owing to its high specificity, sensitivity, and operational simplicity [29,30]. To date, RT-qPCR has been reported for the detection of PoAstV [31,32], PoSaV [33,34], and PoRVA [35]. It is also used for the differentiation of different genotypes of PoNoV [36], and different species of PoRV [37,38]. In addition, multiplex RT-qPCR assays have been established to detect PoSaV/PoNoV [39], PoNoV/PoRVA [40], and PoAstV/PoSaV/PoRVA [28]. However, to date, no multiplex RT-qPCR has been reported for the simultaneous detection of PoAstV, PoSaV, PoNoV, and PoRVA. In this study, specific primers and probes are designed to target the ORF1 genes of PoAstV, PoSaV, and PoNoV, and the VP6 gene of PoRVA, and a quadruplex RT-qPCR assay is established for the simultaneous detection of these four viruses.

## 2. Materials and Methods

### 2.1. Reference Strains

The following vaccine strains were purchased from Keqian Biology Co., Ltd. (Wuhan, China): classical swine fever virus (CSFV, C strain), food-and-mouth disease virus (FMDV, O/Mya98/XJ/2010 strain), porcine pseudorabies virus (PRV, Bartha-K61 strain), porcine reproductive and respiratory syndrome virus (PRRSV, TJM-F92 strain), swine influenza virus (SIV, TJ strain), transmissible gastroenteritis virus (TGEV, H strain), porcine epidemic diarrheal virus (PEDV, CV777 strain), PoRVA (NX strain), and porcine circovirus type 2 (PCV2, SX07 strain).

The following positive clinical samples were provided by our laboratory: PoAstV, PoSaV, PoNoV, PoRVA, African swine fever virus (ASFV), and porcine deltacoronavirus (PDCoV).

### 2.2. Clinical Samples

Between January 2023 and September 2024, 1578 clinical diarrhea feces samples were collected from diarrheic pigs on 44 pig farms in 14 cities of Guangxi province, China. The samples were maintained at ≤4 °C after collection, delivered to the laboratory within 6 h, and stored at −80 °C until use.

### 2.3. Primers and Probes

The reference genome sequences of PoAstV, PoSaV, PoNoV, and PoRVA were downloaded from the NCBI GenBank (https://www.ncbi.nlm.nih.gov/nucleotide/, accessed on 20 October 2022) (Appendix A). The multiple sequence alignments were performed (Appendix A), and four sets of primers and probes for the amplification of the ORF1 gene of PoAstV, PoSaV, and PoNoV, and the VP6 gene of PoRVA were designed (Table 1).

### 2.4. Nucleic Acids

The clinical fecal sample (about 0.3 g) was put into a 2.0 mL EP tube, followed by adding phosphate-buffered saline (pH 7.2, *w*/*v* = 1:4), vortex (1 min), and centrifugation (12,000 rpm, 10 min, 4 °C). The TaKaRa MiniBEST Viral RNA/DNA Extraction Kit Ver.5.0 (Dalian, China) was used to extract the total DNA/RNA from 200 µL supernatant of the fecal sample or vaccine solution.

### 2.5. Standard Plasmid Constructs

The recombinant standard plasmid constructs were constructed according to the method described by Liu et al. [41] with minor modification. The total nucleic acids were reverse transcribed to cDNA, and used to amplify target fragments of PoAstV, PoSaV, PoNoV, and PoRVA using PCR. The products were used to construct the recombinant standard plasmids per the reference method [41]. Finally, the obtained recombinant plasmid constructs were validated by sequencing, and designated as p-PoAstV, p-PoSaV, p-PoNoV, and p-PoRVA, respectively. Their concentrations were determined through measuring the optical density (OD) values at 260 nm and 280 nm with a Thermo Fisher spectrophotometer (Waltham, MA, USA), and calculating per the formula:$$\mathrm{plasmid}\;(\mathrm{copies}/\text{µ}L)=\frac{\left(6.02\times 10^{23}\right)\times (X\;ng/µL\times 10^{-9})}{plasmid\;length\;(bp)\times 660}.$$

### 2.6. Reaction Parameters

Using a mixture of four standard plasmid constructs as the template, the quadruplex RT-qPCR tests were conducted to ascertain the optimal reaction conditions, through varying annealing temperatures (55–60 °C), primer and probe concentrations (20 pmol/µL, 0.2–0.4 µL), and reaction cycles (25–45 cycles). All amplification reactions were conducted using an ABI QuantStudio™ 5 Real-Time System (Carlsbad, CA, USA). The fluorescence signals were automatically recorded for each cycle. Through the analysis of the parameters of maximum ∆Rn and minimum cycling threshold (Ct), the optimal reaction conditions were obtained.

### 2.7. Standard Curves

Four standard plasmid constructs p-PoAstV, p-PoSaV, p-PoNoV, and p-PoRVA were mixed at a ratio of 1:1:1:1, and used as the template for amplification. The concentrations of standard plasmid constructs ranging from $10^{8}$ to $10^{2}$ copies/µL (final reaction concentrations from $10^{7}$ to $10^{1}$ copies/µL) were used to generate the standard curves.

### 2.8. Analytical Specificity

The developed assay’s specificity was analyzed using the viral nucleic acids from PoAstV (astrovirus 3 strain), PoRVA (G5 strain), PoNoV (GII.11 strain), PoSaV (G3 strain), ASFV, FMDV, CSFV, PRRSV, PRV, SIV, PEDV, TGEV, PDCoV, and PCV2 as templates. The p-PoAstV, p-PoSaV, p-PoNoV, and p-PoRVA mixture, negative fecal samples, and distilled water were used as controls.

### 2.9. Analytical Sensitivity

The sensitivity of the developed assay was analyzed using the p-PoAstV, p-PoSaV, p-PoNoV, and p-PoRVA combination with an equal ratio from $10^{8}$ to $10^{0}$ copies/µL ($10^{7}$ to $10^{-1}$ copies/µL as final reaction concentrations). The limit of detection (LOD) of the assay was determined via probit regression analysis.

### 2.10. Repeatability Analysis

The repeatability of the developed assay was analyzed using the p-PoAstV, p-PoSaV, p-PoNoV, and p-PoRVA combination with concentrations of $10^{8}$, $10^{6}$, $10^{4}$ copies/µL (final reaction concentrations of $10^{7}$, $10^{5}$, $10^{3}$ copies/µL) as templates. The tests were performed in triplicate for intra-assay coefficients of variation (CVs) and on three separate days for inter-assay CVs.

### 2.11. Assessment Using Clinical Samples

To assess the developed assay’s applicability, total viral nucleic acids were extracted from the 1578 clinical fecal samples from Guangxi province and tested for PoAstV, PoSaV, PoNoV, and PoRVA using the established quadruplex RT-qPCR. Furthermore, the 1578 samples were assessed using the RT-qPCR established by Padmanabhan et al. for the detection of PoAstV [31], by Shen et al. for the detection of PoSaV [33], by Daseul et al. for the detection of PoNoV [36], and by Shi et al. for the detection of PoRVA [37]. The positivity rates of PoAstV, PoSaV, PoNoV, and PoRVA obtained from the developed assay and the reference assays were compared, and their coincidence rates were assessed.

## 3. Results

### 3.1. Construction of the Standard Plasmids

The total RNAs were extracted from the positive clinical samples of PoAstV, PoSaV, PoNoV, and PoRVA, respectively, reverse transcribed to cDNA, and taken as templates to amplify the ORF1 gene fragments of PoAstV, PoSaV, and PoNoV, and the VP6 gene fragment of PoRVA using PCR. The amplified fragments were used to construct the recombinant standard plasmids, respectively. Finally, the obtained standard plasmid constructs, named p-PoAstV, p-PoSaV, p-PoNoV, and p-PoRVA, had original concentrations of 3.45 × $10^{10}$, 3.21 × $10^{10}$, 1.67 × $10^{10},$ and 2.89 × $10^{10}$ copies/µL, respectively. They were standardized to a concentration of 1.00 × $10^{10}$ copies/µL and conserved at −80 °C until use.

### 3.2. Attainment of the Reaction Conditions

The optimal reaction conditions for quadruplex RT-qPCR were attained by optimizing the annealing temperature, primer and probe concentration, and reaction cycle. Finally, the entire reaction volume was 20 µL, and the ingredients of the amplification system are presented in Table 2. The samples underwent amplification using a One-Step PrimeScript™ RT-PCR Kit (TaKaRa, Dalian, China). All primers and probes were diluted to 20 pmol/μL and used to prepare the reaction systems. The reaction protocol consisted of 42 °C for 5 min, 95 °C for 10 s, followed by 40 cycles of 95 °C for 5 s and 58 °C for 33 s. Fluorescence signals were gathered at the end of each cycle, and the sample with a Ct value ≤ 36 was judged as the positive sample.

### 3.3. Generation of the Standard Curves

The standard curves were generated using a mixture of four standard plasmid constructs at a concentration from $10^{8}$ to $10^{2}$ copies/µL (final reaction concentration: $10^{7}$ to $10^{1}$ copies/µL) as templates. The results indicated that PoAstV (slope = −3.289, R^2^ = 0.999, Eff% = 101.406), PoSaV (slope = −3.289, R^2^ = 0.998, Eff% = 101.411), PoNoV (slope = −3.299, R^2^ = 0.998, Eff% = 100.953), and PoRVA (slope = −3.377, R^2^ = 1, Eff% = 100.4) had good correlation coefficients (R^2^ ≥ 0.998) and amplification efficiencies (E) (Figure 1).

### 3.4. Specificity

The total nucleic acids of PoAstV, PoSaV, PoNoV, PoRVA, ASFV, FMDV, CSFV, PRRSV, PRV, SIV, PEDV, TGEV, PDCoV, and PCV2 from positive clinical samples or vaccine solutions were used to assess the developed assay’s specificity. The results indicated that the assay produced specific amplification curves only for PoAstV, PoSaV, PoNoV, and PoRVA (Figure 2), while all other control viruses were undetectable by any fluorescent signal, indicating good specificity of the assay.

### 3.5. Sensitivity

The four standard plasmid constructs p-PoAstV, p-PoSaV, p-PoNoV, and p-PoRVA, were mixed at equal volumes (1:1:1:1) and then diluted 10-fold serially, and the sensitivity of the assay was evaluated using the plasmid mixtures with the final reaction concentrations of $10^{7}$ to $10^{-1}$ copies/µL. The results indicated that the LODs were determined to be $10^{1}$ copies/μL for all four plasmid constructs (Figure 3).

The LODs were also evaluated by probit regression analysis using four serial dilutions of 500, 250, 125, and 62.5 copies/reaction of p-PoAstV, p-PoSaV, p-PoNoV, and p-PoRVA (Table 3). The results indicated that the LODs of p-PoAstV, p-PoSaV, p-PoNoV, and p-PoRVA were confirmed as 138.001 (95% confidence interval (CI) of 126.235–160.439), 135.167 (95% CI of 123.470–156.146), 140.732 (95% CI of 128.740–165.327), and 132.199 (95% CI of 120.391–152.436), respectively (Figure 4).

### 3.6. Repeatability

The repeatability of the developed assay was assessed using a combination of p-PoAstV, p-PoSaV, p-PoNoV, and p-PoRVA standard plasmid constructs at final concentrations of $10^{7}$, $10^{5}$, and $10^{3}$ copies/μL, respectively. The results indicated that the intra- and the inter-assay CVs were 0.09−1.24% and 0.08−1.11%, respectively (Table 4).

### 3.7. Assessment Results of Clinical Samples

The established quadruplex RT-qPCR assay was used to analyze the 1578 fecal samples from 44 pig farms of 14 cities in Guangxi province. The results indicated that the positivity rates for PoAstV, PoSaV, PoNoV, and PoRVA were 35.93% (567/1578), 8.37% (132/1578), 2.98% (47/1578), and 14.32% (226/1578), respectively. The PoAstV + PoRVA co-infection had the highest positivity rate of 7.67% (121/1578) among dual infections. In addition, 2.28% (36/1578) of the samples exhibited triple infection, and 0.38% (6/1578) of the samples exhibited quadruple infection of PoAstV + PoSaV + PoNoV + PoRVA (Table 5).

Meanwhile, the 1578 diarrhea samples were examined with the methods reported for the detection of PoAstV [31], PoSaV [33], PoNoV [36], and PoRVA [37]. The positivity rates of PoAstV, PoSaV, PoNoV, and PoRVA were 35.61% (562/1578), 8.24% (130/1578), 2.98% (47/1578), and 14.13% (223/1578), respectively. In comparison to the reference methods, the developed assay demonstrated a clinical sensitivity and specificity of 99.64% and 99.31% for PoAstV, 99.23% and 99.79% for PoSaV, 100.00% and 100.00% for PoNoV, and 99.55% and 99.70% for PoRVA, respectively (Table 6). These methods had coincidence rates exceeding 99.43% (Table 7).

## 4. Discussion

Since its first discovery more than 40 years ago, PoAstV has been widely reported worldwide. PoAstV is a diarrhea-associated pathogen and can infect pigs of all ages, with the highest prevalence rate in lactating piglets [42]. PoAstV is widely distributed in China, and has been reported in many provinces [43]. One study found that its infection rate reached 56.4% in Guangxi province from 2013 to 2015 [44]. However, there has been no effective vaccine for PoAstV until now. PoSaV can infect pigs of all ages and cause diarrhea in piglets in particular, and it has been found that the prevalence rate of PoSaV is similar in healthy and diarrheic pigs, and the prevalence rate is significantly higher in weaned pigs than in finishing pigs [45]. According to Liu et al. [46], the prevalence rate of PoSaV in Yunnan province in China in 2020 was 35.2% (71/202), while the vast majority of cases were group GIII. No specific treatment is available for PoSaV infection in large-scale swine farms, and comprehensive control measures are mainly adopted to control the infection. PoNoV, which belongs to the same family of *Caliciviridae* as PoSaV, infects only adult pigs and some infected pigs do not show clinical signs [47]. Although PoNoV is frequently detected in fecal samples from asymptomatic fattening pigs, and most cases of infection alone do not cause diarrhea in pigs, PoNoV has played an important role in the evolution of *Caliciviridae* viruses, and it has been found that PoNoV circulates in healthy adult pigs [48]. Similarly, there is no vaccine and no specific drug for treating PoNoV, and its prevention and control are mainly based on improved feeding management and hygiene measures. PoRVA is a typical enterovirus that causes diarrhea in piglets. In China, PoRVA is often co-infected with PEDV [49]. A study analyzed 594 samples from 35 pig farms in eastern China during 2017–2019 and revealed a positivity rate of PoRVA of 16.8% [50]. PoRVA first appeared in Australia in 1975 [26]. Subsequently, PoRVAs, including G3, G4, G5, G9, and G11 genotypes, have been found in pigs worldwide [51]. Currently, no specific drug can be used for the treatment of PoRVA infection, so the prevention and control strategy for PoRVA is still based on vaccination. However, since vaccination cannot provide complete protection, PoRVA is still prevalent in many countries around the world [21,24,25,26,27,52]. Due to the absence of effective medications or vaccinations, the epidemiological status of these four illnesses remains critical in many countries, including China, which necessitates ongoing prevention and control measures.

Pigs infected with PoAstV, PoSaV, PoNoV, and PoRVA usually show similar symptoms of diarrhea, vomiting, and dehydration, which makes it impossible to correctly distinguish and diagnose these diseases depending only on clinical signs. Even worse, mixed infection of these diarrheal viruses usually occurs [5,6,28,32,40,45,48], which further increases the difficulty of differential diagnosis. The techniques employed to identify viruses include viral isolation, electron microscopy, serological testing, and nucleic acid detection, of which nucleic acid-based assays and virus isolation are widely used. However, these methods usually have certain drawbacks; they are time-consuming, have complicated operations, are susceptible to contamination, and possess insufficient sensitivity. qPCR has emerged as a widely utilized test in laboratories due to its elevated sensitivity, excellent specificity, operational simplicity, high throughput capacity, and its ability to function in a closed environment, minimizing contamination risks [29,30]. Singleplex RT-qPCR assays have been established for the detection of PoAstV, PoSaV, PoNoV, and PoRVA [31,32,33,34,35,36,37,38], and multiplex RT-qPCRs have been established for the simultaneous detection of two pathogens of PoSaV and PoNoV [39,40]. Nonetheless, no multiplex RT-qPCR has been reported for the simultaneous detection of PoAstV, PoSaV, PoNoV, and PoRVA. To establish the quadruplex RT-qPCR, the ORF1 genes of PoAstV, PoSaV, and PoNoV, and the VP6 gene of PoRVA were selected as the target regions, then four pairs of specific primers and probes were designed, the recombinant standard plasmids were constructed, and the optimal reaction conditions were optimized. Finally, a quadruplex RT-qPCR assay for the simultaneous detection and differentiation of PoAstV, PoSaV, PoNoV, and PoRVA was successfully developed. The method had low LODs of 138.001, 135.167, 140.732, and 132.199 copies/reaction for PoAstV, PoSaV, PoNoV, and PoRVA, respectively, with intra- and inter-assay CVs of less than 1.24%. Except for PoAstV, PoSaV, PoNoV, and PoRVA, the other 10 control porcine viruses did not produce any positive fluorescent signal. The developed assay is easy to operate, has good specificity and high sensitivity, and obtained the test results in a short time. The assay validated its applicability through testing 1578 samples from 14 cities in Guangxi province, with higher than 99.43% agreement with the reported reference RT-qPCR assays [31,33,36,37].

In the 1578 tested clinical samples, all four viruses of PoAstV, PoSaV, PoNoV, and PoRVA were found, with positivity rates of 35.93% (567/1578), 8.37% (132/1578), 2.98% (47/1578), and 14.32% (226/1578), respectively. The results showed that these four viruses are still commonly circulating in Guangxi province. The prevalence and co-infections of PoAstV, PoSaV, PoNoV, and PoRVA have also been reported in different countries [5,28,32,53]. The 306 anal swab samples of diarrheic pigs from farms in India showed that the positivity rates of PoAstV and PoRVA were 33.67% (103/306) and 10.6% (32/306), respectively, and their co-infection rate was 4.5% (14/306), with the highest prevalence rates of PoAstV and PoRVA in 3–6-week-old pigs [5]. A total of 280 samples from 28 pig farms in Greece showed positivity rates of 95.4% (267/280) for PoAstV, and 20.4% (57/280) for Caliciviruses [32]. A total of 483 diarrhea samples from northern China were found to be positive for PoAstV, and PoSaV at 35.4% (171/483), and 18.4% (89/483), respectively, and 11.8% (57/483) of the samples were positive for PoAstV and PoSaV [53]. A total of 280 pig fecal samples from Heilongjiang province in China showed positivity rates of PoSaV, PoRVA, and PoAstV of 24.6% (69/280), 4.3% (12/280), and 17.5% (49/280), respectively, and co-infections of 5% for PoSaV + PoRVA+PoAstV, 8% for PoSaV + AstV, 2% for PoRVA + PoAstV, and 1% for PoRVA + PoSaV [28]. These results indicate that these diseases may vary greatly in different countries. This study found that the positivity rate of PoAstV was as high as 35.93%, so the threat of this virus to the pig industry in Guangxi province cannot be ignored. Co-infections were also found, with the highest rate of 7.67% (121/1578) for PoAstV and PoRVA, and quadruple co-infections were found in six samples from Nanning City. Although the PoRVA vaccine has been widely used, the disease is still prevalent in Guangxi province. The high prevalence and co-infection rates of these viruses seriously threaten the production safety of the pig industry. The developed assay in this study provides a technique for the quick, easy, and accurate detection of these four viruses, which is vital to accurately diagnose these diseases.

It is noteworthy that PoAstV, PoSaV, PoNoV, and PoRVA have a wide range of host species, and can acquire the ability of trans-species transmission. They have the potential to be transmitted from animals to humans, showing high zoonotic potential [1,9,46,50,54,55]. The epidemics of these viruses cause serious economic losses to animals, and also pose a great threat to human health. The common prevalence of PoAstV, PoSaV, PoNoV, and PoRVA in different countries suggests that prevention and control measures have to be strengthened for these diseases, which can not only reduce the losses to the pig industry, but also protect humans from the harm of zoonotic diseases.

Due to their unique genetic characteristics, and their origination from different countries and dates, the clinically prevalent strains of PoAstV, PoSaV, PoNoV, and PoRVA show a high degree of genetic diversity [9,11,14,16,18,21,24,43,45,46]. This places high demands on the specificity and sensitivity of the detection methods used for these viruses. In this study, the genome sequences of representative strains of these viruses from different countries and dates were downloaded from NCBI GenBank, and multiple sequence alignments were performed. Then, specific primers and probes were designed targeting the genetic conserved regions so that they could specially detect different genotypes and strains of these viruses. Of course, due to the continuous variation of the genome of prevalent strains, it is necessary to analyze the newest viral sequences and adjust the primer and probe sequences to avoid missing the detection of clinical mutated strains. In addition, the mixed infections of different viruses might decrease the sensitivity of the multiplex RT-qPCR [29,30], which will result in missing some positive samples with very low viral loads. To increase the assay’s sensitivity, the experiments were performed using different combinations of primer and probe concentrations, annealing temperatures, and reaction cycles, and the optimal conditions were finally determined. In this study, after optimizing the reaction conditions using the specific primers and probes, a quadruplex RT-qPCR assay with high specificity and sensitivity was developed for the detection of PoAstV, PoSaV, PoNoV, and PoRVA.

## 5. Conclusions

With its user-friendliness, high sensitivity, and excellent specificity, the developed quadruplex RT-qPCR assay in this study is capable of the simultaneous detection of PoAstV, PoSaV, PoNoV, and PoRVA. The applicability of this assay allows for the fast and precise detection of potential zoonotic swine viruses, which significantly contributes to the control and prevention of these viruses. The analysis of clinical samples reveals that PoAstV, PoSaV, PoNoV, and PoRVA are still prevalent in Guangxi province in China. More attention should be paid to these gastroenteric viruses.

## Figures and Tables

**Figure 1 pathogens-13-01052-f001:**
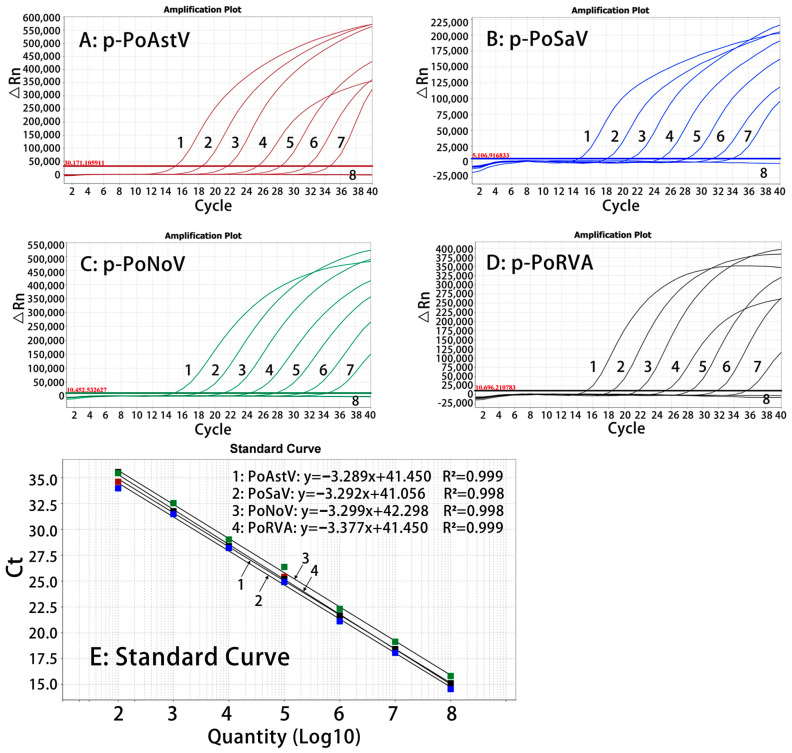
The amplification curves of p-PoAstV (**A**), p-PoSaV (**B**), p-PoNoV (**C**), and p-PoRVA (**D**), and the standard curves (**E**). In (**A**–**D**), 1–7: the concentrations of standard plasmid constructs ranged from $10^{7}$ to $10^{1}$ copies/µL; 8: nuclease-free distilled water. In (**E**), 1–4: the standard curves of p-PoAstV, p-PoSaV, p-PoNoV, and p-PoRVA, respectively.

**Figure 2 pathogens-13-01052-f002:**
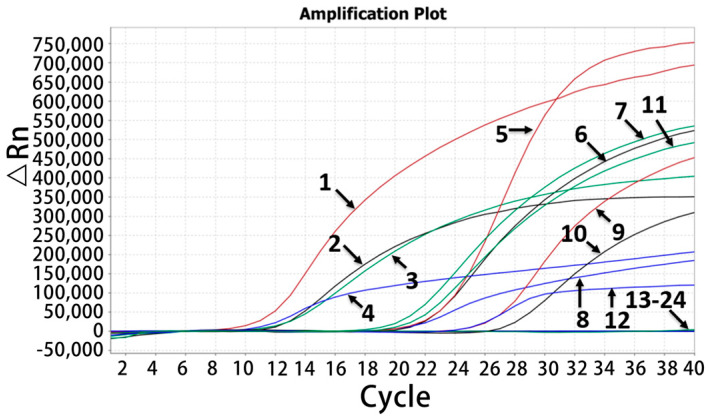
Specificity assessment of the quadruplex RT-qPCR. 1: p-PoAstV; 2: p-PoRVA; 3: p-PoNoV; 4: p-PoSAV; 5–8: the positive clinical samples of PoAstV, PoRVA, PoNoV, and PoSaV; 9: PoAstV (astrovirus 3 strain); 10: PoRVA (G5 strain); 11: PoNoV (GII.11 strain); 12: PoSaV (G3 strain); 13–22: ASFV, FMDV, CSFV, PRRSV, PRV, SIV, PEDV, TGEV, PDCoV, and PCV2; 23: negative fecal sample; 24: distilled water.

**Figure 3 pathogens-13-01052-f003:**
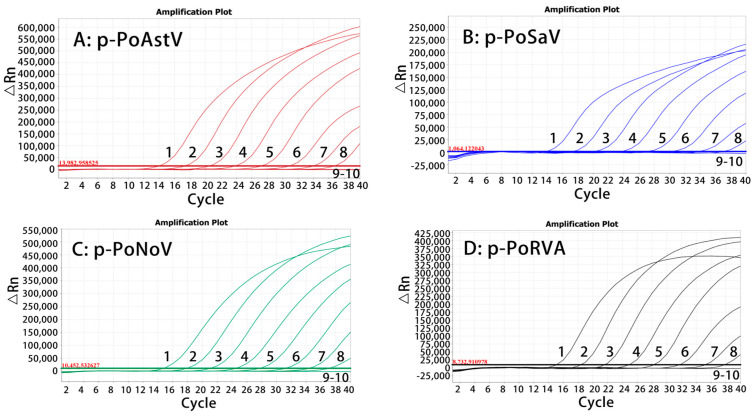
Sensitivity assessment of the quadruplex RT-qPCR. In (**A**–**D**), 1–9: the concentrations of standard plasmid constructs ranged from $10^{7}$ to $10^{-1}$ copies/μL (final reaction concentrations); 10: negative control.

**Figure 4 pathogens-13-01052-f004:**
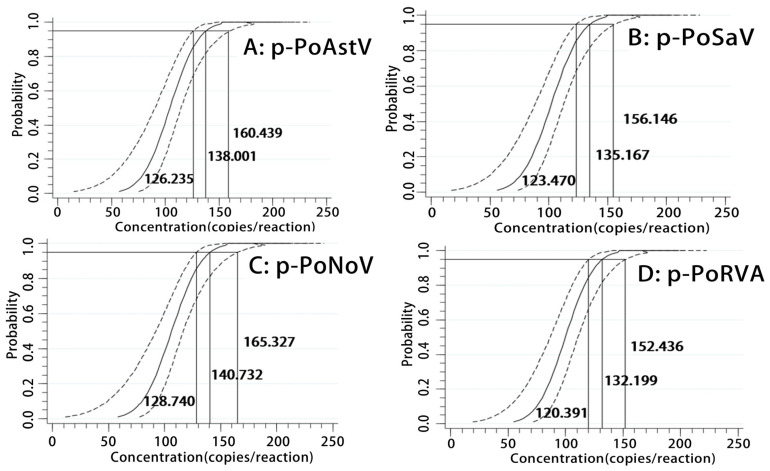
Assessment of sensitivity using probit regression analysis. The p-PoAstV (**A**), p-PoSaV (**B**), p-PoNoV (**C**), and p-PoRVA (**D**) had LODs of 138.001 (95% CI of 126.235–160.439), 135.167 (95% CI of 123.470–156.146), 140.732 (95% CI of 128.740–165.327), and 132.199 (95% CI of 120.391–152.436), respectively.

**Table 1 pathogens-13-01052-t001:** The specific primers and probes.

Primer/Probe	Sequence (5′ → 3′)	Gene	Tm/°C	Size (bp)
PoAstV-F	GATTTACAGTTGGCCCAGA	ORF1	56.2	116
PoAstV-R	GAGTTTCCCATGCAGCG		56.6	
PoAstV-P	ROX-CCCACTGATGAAGAGAAACTCTATGCT-BHQ2		62.3	
PoSaV-F	GGCCAACGCAGTGGCAAC	ORF1	61.6	96
PoSaV-R	ATCACGAACACTTCTGGCTC		58.3	
PoSaV-P	CY5-GCATGGTACGGTGGCACTGACGG-BHQ3		66.4	
PoNoV-F	GGACCTCCTTGCCCCCA	ORF1	59.9	135
PoNoV-R	GACATGGCACAGAGRGTGAT		57.9	
PoNoV-F	VIC-ACATCACAATGGAACTCCTTTGCCC-BHQ1		63.4	
PoRVA-F	TGACACCAGCAGTTGCA	VP6	54.9	99
PoRVA-R	ATTCACAAACTGCAGATTCA		54.1	
PoRVA-P	FAM-AGCACCACCATTTATATTTCATGCTAC-BHQ1		60	

**Table 2 pathogens-13-01052-t002:** The components and the optimal parameters.

Ingredient	Volume (µL)	Final Concentration (nM)
2× One Step RT-PCR Buffer III	10.0	/
Ex Taq HS (5 U/µL)	0.4	/
PrimeScript RT Enzyme Mix II	0.4	/
PoAstV-F	0.2	200
PoAstV-R	0.2	200
PoAstV-P	0.2	200
PoSaV-F	0.2	200
PoSaV-R	0.2	200
PoSaV-P	0.2	200
PoNoV-F	0.4	400
PoNoV-R	0.4	400
PoNoV-P	0.3	300
PoRVA-F	0.3	300
PoRVA-R	0.3	300
PoRVA- P	0.2	200
Total Nucleic Acid	2.0	/
Nuclease-Free Distilled Water	Up to 20.0	/

**Table 3 pathogens-13-01052-t003:** The Ct values and hit rates of the plasmid constructs with serial dilution.

Plasmid Constructs	Copies/Reaction	Number of Samples	Quadruplex RT-qPCR	
			Ct Value (Average)	Hit Rate (%)
p-PoAstV	500	28	34.81	100
	250	28	35.36	100
	125	28	35.86	85.71
	62.5	28	Not Detected	0
p-PoSaV	500	28	34.67	100
	250	28	35.24	100
	125	28	35.83	89.29
	62.5	28	Not Detected	0
p-PoNoV	500	28	34.22	100
	250	28	34.87	100
	125	28	35.45	82.14
	62.5	28	Not Detected	0
p-PoRVA	500	28	34.35	100
	250	28	34.88	100
	125	28	35.36	92.86
	62.5	28	Not Detected	0

**Table 4 pathogens-13-01052-t004:** Repeatability of the quadruplex RT-qPCR.

Plasmid Constructs	Concentration (Copies/μL)	Ct Values of Intra-Assay			Ct Values of Inter-Assay		
		$\bar{X}$	SD	CV (%)	$\bar{X}$	SD	CV (%)
p-PoAstV	10^7^	15.585	0.17	1.08	15.437	0.09	0.57
	10^5^	23.470	0.11	0.45	23.437	0.08	0.34
	10^3^	27.900	0.02	0.09	27.928	0.02	0.08
p-PoSaV	10^7^	17.208	0.11	0.65	17.415	0.12	0.80
	10^5^	25.339	0.31	1.23	25.474	0.26	1.03
	10^3^	29.341	0.11	0.35	29.161	0.13	0.44
p-PoNoV	10^7^	17.185	0.21	1.24	17.272	0.11	0.65
	10^5^	25.042	0.11	0.45	25.442	0.09	0.35
	10^3^	30.014	0.06	0.21	29.641	0.17	0.58
p-PoRVA	10^7^	16.713	0.15	0.92	16.295	0.18	1.11
	10^5^	24.468	0.07	0.28	24.309	0.15	0.62
	10^3^	29.038	0.12	0.41	29.212	0.15	0.50

**Table 5 pathogens-13-01052-t005:** Detection results of the clinical samples.

City		Positive Sample														
	Number	PoAstV	PoSaV	PoNoV	PoRVA	A + S	A + N	A + R	S + N	S + R	N + R	A + S + N	A + S + R	A + N + R	S + N + R	A + S + N + R
Fangchenggang	80	30	0	0	9	0	0	3	0	0	0	0	0	0	0	0
Liuzhou	69	42	5	5	3	2	1	3	0	0	0	0	0	0	0	0
Nanning	162	77	45	20	30	26	12	24	5	8	6	10	6	14	6	6
Chongzuo	45	14	20	0	0	10	0	0	0	0	0	0	0	0	0	0
Yulin	125	44	20	14	0	10	9	0	3	0	0	2	0	0	0	0
Guigang	286	122	27	0	43	6	0	23	0	4	0	0	4	0	0	0
Baise	593	149	15	4	125	14	0	56	0	0	0	0	0	0	0	0
Hezhou	30	6	0	2	3	0	2	3	0	0	0	0	0	0	0	0
Hechi	21	12	0	2	0	0	2	0	0	0	0	0	0	0	0	0
Beihai	20	4	0	0	0	0	0	0	0	0	0	0	0	0	0	0
Laibin	23	8	0	0	0	0	0	0	0	0	0	0	0	0	0	0
Wuzhou	54	30	0	0	10	0	0	7	0	0	0	0	0	0	0	0
Qinzhou	50	23	0	0	3	0	0	2	0	0	0	0	0	0	0	0
Guilin	20	6	0	0	0	0	0	0	0	0	0	0	0	0	0	0
Total (%)	1578	567 (35.93%)	132 (8.37%)	47 (2.98%)	226 (14.32%)	68 (4.31%)	26 (1.65%)	121 (7.67%)	8 (0.51%)	12 (0.76%)	6 (0.38%)	12 (0.76%)	10 (0.63%)	14 (0.89%)	6 (0.38%)	6 (0.38%)

Note: A + S denotes co-infection of PoAstV and PoSaV; A + N denotes co-infection of PoAstV and PoNoV; A + R denotes co-infection of PoAstV and PoRVA; S + N denotes co-infection of PoSaV and PoNoV; S + R denotes co-infection of PoSaV and PoRVA; N + R denotes co-infection of PoNoV and PoRVA; A + S + N denotes co-infection of PoAstV, PoSaV, and PoNoV; A + S + R denotes co-infection of PoAstV, PoSaV, and PoRVA; A + N + R denotes co-infection of PoAstV, PoNoV, and PoRVA; S + N + R denotes co-infection of PoSaV, PoNoV, and PoRVA; A + S + N + R denotes co-infection of PoAstV, PoSaV, PoNoV, and PoRVA.

**Table 6 pathogens-13-01052-t006:** The clinical sensitivity and specificity of the established assay.

The Established Assay		The Reference Assay		Total	Clinical Sensitivity (95% CI)	Clinical Specificity (95% CI)
		Positive	Negative			
PoAstV	Positive	560	7	567	99.64% (98.71–99.90%)	99.31% (98.58–99.67%)
	Negative	2	1009	1011		
	Total	562	1016	1578		
PoSaV	Positive	129	3	132	99.23% (95.77–99.86%)	99.79% (99.50–99.96%)
	Negative	1	1445	1446		
	Total	130	1448	1578		
PoNoV	Positive	47	0	47	100.00% (92.44–100.00%)	100.00% (99.75–100.00%)
	Negative	0	1531	1531		
	Total	47	1531	1578		
PoRVA	Positive	222	4	226	99.55% (97.50–99.92%)	99.70% (99.33–99.90%)
	Negative	1	1351	1352		
	Total	223	1355	1578		

**Table 7 pathogens-13-01052-t007:** Agreements of the detection results using different assays.

Method	Positive Samples			
	PoAstV	PoSaV	PoNoV	PoRVA
The Developed Multiplex RT-qPCR	35.93% (567/1578)	8.37% (132/1578)	2.98% (47/1578)	14.32% (226/1578)
The Reported Reference RT-qPCR	35.61% (562/1578)	8.24% (130/1578)	2.98% (47/1578)	14.13% (223/1578)
Agreements	99.43%	99.75%	100%	99.68%

## Data Availability

Data are contained within the article and Appendix A.

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
