# Peer review of "Development of a Quadruplex RT-qPCR for the Detection of Porcine Astrovirus, Porcine Sapovirus, Porcine Norovirus, and Porcine Rotavirus A"

_pathogens, 2024, doi:10.3390/pathogens13121052_

Round 1

Reviewer 1 Report

Comments and Suggestions for Authors

The manuscript entitled "Development of a quadruplex RT-qPCR for the detection of porcine astrovirus, porcine sapovirus, porcine norovirus, and porcine rotavirus A."  describes a multiplex diagnostic tool for detecting pathogens involved in diarrhea in pigs. Some of which have zoonotic potential.  The work involves the development of the assay and the testing of field samples from a region in China. The methods and presentation of the data is sound, but the text is at time hard to follow. The authors could also discuss pitfall, limitation to implementation in more diverse sets of samples (differences in regional strains, high viral loads in mixed infections, etc).

Bellow a few suggested edits: 

Lines 88-90 (the entire paragraph): Please, eliminate past tense. Same for the following: “Therefore, it had to rely on laboratory testing techniques for differential diagnosis (…)”. The paragraph could be adjusted for clarity and flow.

Line 90: The phrase "The co-infections of them increases" would read more smoothly as "Co-infection by these viruses increases," to improve grammar and readability.

Lines 95-103: The section contains multiple sentences coordinated by semi-colons. I suggest reviewing and simplifying the structure for better readability and flow, potentially breaking up the sentences for clarity.

Lines 381-387: Please revise to improve text and flow.  

Comments on the Quality of English Language

While the more descriptive paragraphs are well written, other parts of the manuscript have some grammar and flow deficiency. 

Author Response

The Cover Letter

November 23, 2024

Dear editor,

Our manuscript has been revised carefully according to the reviewers’ suggestions. The details are as follows.

Reviewer #1

Comments and Suggestions for Authors

The manuscript entitled "Development of a quadruplex RT-qPCR for the detection of porcine astrovirus, porcine sapovirus, porcine norovirus, and porcine rotavirus A."  describes a multiplex diagnostic tool for detecting pathogens involved in diarrhea in pigs. Some of which have zoonotic potential.  The work involves the development of the assay and the testing of field samples from a region in China. The methods and presentation of the data is sound, but the text is at time hard to follow.

  1. The authors could also discuss pitfall, limitation to implementation in more diverse sets of samples (differences in regional strains, high viral loads in mixed infections, etc.).

Response: We agree to the reviewer’s suggestion. The pitfall, and the limitation to implementation in more diverse sets of samples have been discussed in the Discussion section. Please see Lines 409-427 in the revised manuscript.

Bellow a few suggested edits: 

  1. Lines 88-90 (the entire paragraph): Please, eliminate past tense. Same for the following: “Therefore, it had to rely on laboratory testing techniques for differential diagnosis (…)”. The paragraph could be adjusted for clarity and flow.

Response: We agree to the reviewer’s suggestion. The past tense has been eliminated. In addition, the entire paragraph has been revised carefully. Please see Lines 91-107 in the revised manuscript.

  1. Line 90: The phrase “The co-infections of them increases” would read more smoothly as “Co-infection by these viruses increases,” to improve grammar and readability.

Response: We agree to the reviewer’s suggestion. “The co-infections of them increases” has been changed to “Co-infection by these viruses increases,”. Please see Lines 93-94 in the revised manuscript.

  1. Lines 95-103: The section contains multiple sentences coordinated by semi-colons. I suggest reviewing and simplifying the structure for better readability and flow, potentially breaking up the sentences for clarity.

Response: We agree to the reviewer’s suggestion. These sentences have been revised. Please see Lines 98-107 in the revised manuscript.

  1. Lines 381-387: Please revise to improve text and flow.  

Response: We agree to the reviewer’s suggestion. The entire paragraph has been revised to improve text and flow. Please see Lines 400-408 in the revised manuscript.

 Comments on the Quality of English Language

  1. While the more descriptive paragraphs are well written, other parts of the manuscript have some grammar and flow deficiency. 

Response: We agree to the reviewer’s suggestion. The whole manuscript has been revised carefully to improve the quality of the manuscript. Please see the revised manuscript.

Reviewer #2

Comments and Suggestions for Authors

  1. The developed quadruplex RT-qPCR assay in this study showed high specificity, sensitivity and repeatability. The objective of study - development of a quadruplex RT-qPCR for detection of porcine astrovirus, sapovirus, norovirus and porcine rotavirus A was fulfilled.

Response: Thanks very much for the reviewer's affirmation of this manuscript.

Reviewer #3

Comments and Suggestions for Authors

Livestock farming, including swine farming, involves rearing animals for different purposes and has tremendously intensified in the last decades. Despite the different advantages that livestock farming brings, there are also some disadvantages, including the increasing animal welfare, alongside environmental, and health concerns due to the high density of the farming itself.

Apart from concerns about the quality of life of the animals raised in such conditions, experts have also expressed concerns regarding biosecurity due to eventual disease outbreaks in farming facilities. Indeed, the risk of massive losses increases as they would be more difficult to contain. Not to mention that some disease-causing agents (i.e., viruses) might display zoonotic potential.

Among the most common pig farming infections there are those brought by porcine diarrhea viruses.

Remarkably, different diarrhea viruses might all display very similar symptoms thus making it hard to distinguish among them. The latter issue is quite important because the detection of the disease-causing agent enables the most appropriate treatment approach.

In the manuscript titled "Development of a quadruplex RT-qPCR for the detection of porcine astrovirus, porcine sapovirus, porcine norovisrus, and porcine rotavirus A" He J. and coworkers set up the reaction conditions, the specificity and sensitivity of a novel quadruplex RT-qPCR test for the detection of porcine astro-, sapo-, noro-, and rota-virus. Eventually, once these parameters were set up the authors assayed more than 1,500 foecal samples from pig farms in the Chinese province of Guangxi.

Overall, the current manuscript closely follows the methodological approach used in others, nonetheless offering the novelty of the simultaneous detection of 4 different kinds of diarrhea associated viruses.

No major issues are detected, nonetheless before publication the authors are asked to make a couple of amendments, shortly below discussed.

  1. Page 3: why numbering the first Table as "Table 7", and then go (see page 5) to "Table 1"? To the reviewer, the rationale behind is not clear and just confuses the readers.

Response: We agree to the reviewer’s suggestion. The first Table as "Table 7" is a mistake. "Table 7" has been changed to "Table 1". Other tables have been changed sequentially to Table 2, 3, 4, 5, 6 and 7. Please see Table 1-7 in the revised manuscript.

  1. Page 5: Figure 1E: the numbers 1, 2, 3, and 4 refer to what? The standard curves are color-labeled. Please clarify the inconsistency.

Response: We agree to the reviewer’s suggestion. The information on the numbers 1, 2, 3, and 4 in Figure 1E has been added in the revised manuscript. Please see Figure 1E and Lines 223-226 in the revised manuscript.

  1. Page 10: "Table 4" What does represent the last row of the table? The authors are asked to elucidate the issue.

Response: We agree to the reviewer’s suggestion. The data in the last row of the table represent the percentage of positive samples detected by the developed quadruplex RT-qPCR assay. Please see Table 5 in the revised manuscript.

  1. Sometimes it seems that the manuscript language is redundant (e.g., section "Discussion" page 11 lines 294-303) and not always clear (e.g., page 13, line 384: potential virus outbreaks surely cause economic damage, but the latter concern the breeders and not animals, which conversely face other kinds, namely health problems).

Response: We agree to the reviewer’s suggestion. The contents in the previous Lines 294-303 and 384 have been revised carefully. Please see Lines 310-319 and Lines 403-404 in the revised manuscript.

In addition, the whole manuscript has been revised carefully to improve the quality of the manuscript. Please see the revised manuscript.

  1. A few typos are scattered throughout the main text (e.g., lines 98, 126, etc…).

Response: We agree to the reviewer’s suggestion. The contents in the previous Lines 98 and 126 have been revised carefully. Please see Lines 101-103 and Lines 134-136 in the revised manuscript.

In addition, the whole manuscript has been revised carefully to improve the quality of the manuscript. Please see the revised manuscript.

Best regards,

Kaichuang Shi

Reviewer 2 Report

Comments and Suggestions for Authors

The developed quadruplex RT-qPCR assay in this study showed high specificity, sensitivity and repeatibility. The objective of study - development of a quadruplex RT-qPCR for detection of porcine astrovirus, sapovirus, norovirus and porcine rotavirus A was fulfilled.

Author Response

(The authors gave the same response as above.)

Reviewer 3 Report

Comments and Suggestions for Authors

Livestock farming, including swine farming, involves rearing animals for different purposes and has tremendously intensified in the last decades. Despite the different advantages that livestock farming brings, there are also some disadvantages, including the increasing animal welfare, alongside environmental, and health concerns due to the high density of the farming itself.

Apart from concerns about the quality of life of the animals raised in such conditions, experts have also expressed concerns regarding biosecurity due to eventual disease outbreaks in farming facilities. Indeed the risk of massive losses increases as they would be more difficult to contain. Not to mention that some disease-causing agents (i.e., viruses) might display zoonotic potential.

Among the most common pig farming infections there are those brought by porcine diarrhea viruses.

Remarkably, different diarrhea viruses might all display very similar symptoms thus making it hard to distinguish among them. The latter issue is quite important because the detection of the disease-causing agent enables the most appropriate treatment approach.

In the manuscript titled "Development of a quadruplex RT-qPCR for the detection of porcine astrovirus, porcine sapovirus, porcine norovisrus, and porcine rotavirus A" He J. and coworkers set up the reaction conditions, the specificity and sensitivity of a novel quadruplex RT-qPCR test for the detection of porcine astro-, sapo-, noro-, and rota-virus. Eventually, once these parameters were set up the authors assayed more than 1,500 foecal samples from pig farms in the Chinese province of Guangxi.

Overall, the current manuscript closely follows the methodological approach used in others, nonetheless offering the novelty of the simultaneous detection of 4 different kinds of diarrhea associated viruses.

No major issues are detected, nonetheless before publication the authors are asked to make a couple of amendments, shortly below discussed.

  1. Page 3: why numbering the first Table as "Table 7", and then go (see page 5) to "Table 1"? To the reviewer, the rationale behind is not clear and just confuses the readers.
  2. Page 5: Figure 1E: the numbers 1, 2, 3, and 4 refer to what? The standard curves are color-labeled. Please clarify the inconsistency. 
  3. Page 10: "Table 4" What does represent the last row of the table? The authors are asked to elucidate the issue.
  4. Sometimes it seems that the manuscript language is redundant (e.g., section "Discussion" page 11 lines 294-303) and not always clear (e.g., page 13, line 384: potential virus outbreaks surely cause economic damage, but the latter concern the breeders and not animals, which conversely face other kinds, namely health problems.
  5. A few typos are scattered throughout the main text (e.g., lines 98, 126, etc…).

Author Response

(The authors gave the same response as above.)
